# The Adaptive Alternation of Intestinal Microbiota and Regulation of Host Genes Jointly Promote Pigs to Digest Appropriate High-Fiber Diets

**DOI:** 10.3390/ani14142076

**Published:** 2024-07-16

**Authors:** Yunchao Zhang, Hui Li, Bengao Li, Jiayi He, Chen Peng, Yanshe Xie, Guiqing Huang, Pengju Zhao, Zhengguang Wang

**Affiliations:** 1Hainan Institute, Zhejiang University, Sanya 572000, China; 22117077@zju.edu.cn (Y.Z.); 22017074@zju.edu.cn (J.H.); 22217072@zju.edu.cn (C.P.); 12217021@zju.edu.cn (Y.X.); 22217094@zju.edu.cn (G.H.); zhaopengju2014@zju.edu.cn (P.Z.); 2College of Animal Sciences, Zhejiang University, Hangzhou 310000, China; 3Long Jian Animal Husbandry Company, Haikou 570100, China; 18668180372@163.com (H.L.); 3170100171@zju.edu.cn (B.L.)

**Keywords:** fibrous diets, gut microbiome, roughage tolerance, Tunchang pigs, transcriptome

## Abstract

**Simple Summary:**

Tunchang pigs are known for their good performance on roughage tolerance, which may alleviate the conflict between humans and livestock for grain resources. This study observed that Tunchang pigs can adapt to a high-fiber diet containing 5% crude fiber. The adaptive alterations in intestinal microbiota and the regulation of genes collectively facilitated the digestion of high-fiber diets in the host. These findings provide insight into the roughage tolerance of Tunchang pigs, and contribute to further exploration of host–microbiota interactions.

**Abstract:**

Although studies have revealed the significant impact of dietary fiber on growth performance and nutrient digestibility, the specific characteristics of the intestinal microbiota and gene regulation in pigs capable of digesting high-fiber diets remained unclear. To investigate the traits associated with roughage tolerance in the Chinese indigenous pig breed, we conducted comparative analysis of growth performance, apparent fiber digestibility, intestinal microbiota, SCFA concentrations and intestinal transcriptome in Tunchang pigs, feeding them diets with different wheat bran levels. The results indicated that the growth performance of Tunchang pigs was not significantly impacted, and the apparent total tract digestibility of crude fiber was significantly improved with increasing dietary fiber content. High-fiber diets altered the diversity of intestinal microbiota, and increased the relative abundance of *Prevotella*, *CF231*, as well as the concentrations of isobutyrate, valerate and isovalerate. The LDA analysis identified potential microbial biomarkers that could be associated with roughage tolerance, such as *Prevotella stercorea*, and *Eubacterium biforme*. In addition, appropriate high-fiber diets containing 4.34% crude fiber upregulated the mRNA expressions of PYY, AQP8, and SLC5A8, while downregulating the mRNA expressions of CKM and CNN1.This indicated that appropriate high-fiber diets may inhibit intestine motility and increase the absorption of water and SCFAs.

## 1. Introduction

The rising demand for feed grain exacerbates the conflict between humans and livestock for grain resources. Therefore, growing attention has been paid to the utilization of agricultural byproducts as feedstuff, such as rice bran and wheat bran, while previous studies found that the addition of fiber-rich ingredients can inhibit nutrient digestibility and reduce feed intake in growing pigs [1,2,3]. These studies have revealed the significant impact of dietary fiber on growth performance and nutrient digestibility. However, there are certain limitations, as they do not investigate the underlying interactions.

Trillions of microorganisms colonizing the host’s large intestine form a complex microbial ecosystem, which is critical for nutrient absorption [4]. The digestive enzymes secreted by the gastrointestinal tract of the pig cannot degrade dietary fiber. Therefore, nondigestible dietary fiber is fermented in the large intestine to support microbial growth. In return, the microbiota generate multiple metabolites, among which short-chain fatty acids (SCFAs) are primary [5]. SCFAs serve as energy substrates, mitigate inflammation, and regulate satiety [6]. A recent study found that the growth performance of weaned piglets fed dietary corn bran or wheat bran improved due to alterations in intestinal microbiota and enhanced butyrate production [7]. Many studies have focused on the impact of dietary fiber on intestinal microbiota and microbial metabolites, overlooking the regulation of dietary fiber on host gene expression, which influences the host’s physiological activities.

Chinese indigenous pig breeds have demonstrated good performance in roughage tolerance [8,9,10], which may be related to intestinal microbiota and host–microbiota interactions [11]. Tunchang pigs, a subpopulation of Hainan pigs, thrive in an environment similar to Southeast Asia, characterized by a mild climate and abundant rainfall. These pigs gradually developed resistance to roughage through ample supplementation of green feed [12]. However, the roughage tolerance of Tunchang pigs remains unknown, and there is a lack of studies investigating the underlying host–microbiota interactions. The present study aimed to investigate the interaction between intestinal microbiota and gene expression in response to feeding high-fiber diets to Tunchang pigs through 16S rRNA gene sequencing and transcriptome sequencing. It was hypothesized that Tunchang pigs would perform well in roughage tolerance and that dietary fiber would affect the composition and function of cecal and colonic microbiota, involving the regulation of intestinal gene expression.

## 2. Materials and Methods

### 2.1. Animals, Diets and Sampling

A total of 24 Tunchang female pigs (day 180) with body weights of 43.71 ± 2.89 kg were randomly allotted into three groups: A, B and C group. Each pig was considered a replicate, resulting in 8 replicates per group. Group A was fed a basal diet, while Group B and C were provided with basal diets supplemented with 8% and 16% wheat bran, respectively. The basal diet was formulated to meet the nutrient requirements of the NRC. According to the “unique difference principle”, the dietary fiber level in each group was adjusted by manipulating the wheat bran content. Additionally, the contents of corn, soybean meal, grease meal, and fish meal were slightly adjusted to ensure the measured values of crude protein (CP) and digestible energy (DE) were nearly identical (Table 1). All pigs had ad libitum access to their diets and clean water during the 111-day experiment. Feed consumption was recorded daily to calculate the average daily feed intake (ADFI). Each pig was weighed at the beginning and end of the trial to determine the average daily gain (ADG).

At the end of the trial (day 111), fresh fecal samples were collected to determine nutrient and fiber digestibility. Five pigs from each group were taken to a slaughterhouse, euthanized by electrical stunning and exsanguination after approximately 12 h of fasting. Digesta from the cecum and colon were collected for 16S rRNA gene sequencing and SCFAs analysis. Sterile ophthalmic scissors were used to excise the cecal and colonic segments for transcriptome sequencing.

### 2.2. Chemical Analysis

The apparent total tract digestibility (ATTD) was determined by using acid-insoluble ash (AIA) as an endogenous indicator. Diets and fecal samples were analyzed for DE, CP, CF and AIA according to the Association of Official Analytical Chemists procedures (AOAC) [13]. ATTD was calculated as follows:ATTD (%) = 100% × (1 − (NC_F_ × AIA_D_)/(NC_D_ × AIA_F_))
where NC_F_ is the nutrient content in feces; AIA_D_ is the content of AIA in diets; NC_D_ is the nutrient content in diets; and AIA_F_ is the content of AIA in feces.

Cecal and colonic SCFA concentrations were determined by a gas chromatography– tandem mass spectrometry platform at Personal Biotechnology Co., Ltd. (Shanghai, China), according to a previous study [14].

### 2.3. 16S rRNA Gene Sequencing

DNA extraction and 16S rRNA gene sequencing were conducted at Personal Bio Inc. (Shanghai, China). Microbial DNA was extracted from the digesta of cecum and colon using a DNeasy PowerSoil Kit (QIAGEN, Hilden, Germany,) according to the manufacturer’s protocols. The V3-V4 hypervariable regions of the bacteria 16S rRNA gene were amplified with primers 338F and 806R by the T100 PCR system (Bio-Rad, Hercules, CA, USA). Purified amplicons were pooled in equimolar paired-end sequencing performed on an Illumina MiSeq platform (Illumina, San Diego, CA, USA), according to the standard protocols [15]. Raw sequenced reads were processed to obtain high-quality clean reads as described by a previous study [16]. Clean reads were demultiplexed and constructed to an amplicon sequence variant (ASV) feature table using the QIIME ‘DADA2’ package [17]. Greengenes database (Release 13.8) [18] was used to predict taxonomic assignments for each ASV sequence. Results were then exported for further analysis in R software (v4.1.0).

### 2.4. Transcriptome Sequencing

Total RNA was extracted from all intestinal segment samples using a Total Kit Ⅱ (OMEGA, Norcross, GA, USA). The TruSeq RNA Library Prep Kit v2 (Illumina, San Diego, CA, USA) was used to construct cDNA libraries according to the manufacturer’s instructions. After RNA extraction, purification, library construction, and PCR, the cDNA libraries were subjected to paired-end sequencing using Illumina NovaSeq 6000 sequencing platform at Personal Bio Inc. (Shanghai, China). The raw data downloaded from platform contains some adapter-contaminated and low-quality reads, which may interfere with subsequent data analysis. Cutadapt (v1.2.1) was used to further filter the raw data to obtain clean data [19]. The clean data was aligned to the reference genome (Sus scrofa 11.1) using the improved BWT algorithm with the HISAT2 software, which offers higher speed and lower resource consumption [20]. The “DESeq2” package in R software (v4.1.0) was used to calculate the fold change in expression between two groups and q-values for multiple testing. Differential expression genes (DEGs) were analyzed and filtered based on the criteria: |log2FoldChange| > 1, and a significant q-value < 0.05.

### 2.5. Quantitative Real-Time PCR

Total RNA was isolated from cecal tissue by using SPARKeasy Plus Universal Kit (SparkJade Bio Inc., Shandong, China), according to the manufacturer’s protocols. Concentration and purity of RNA samples were determined on a NanoDrop 2000 spectrophotometer (Thermo Scientific, Waltham, MA, USA). Single-stranded cDNA was synthesized from 1 μg of total RNA by using the SPARKscript RT Plus Kit (AG0304; SparkJade Bio Inc., Shandong, China), according to the manufacturer’s protocols. Quantitative real-time PCR was carried out with the use of 2 × SYBR Green qPCR Mix (AH0104; SparkJade Bio Inc., Shandong, China). The reaction system of quantitative real-time PCR was 10 μL including 5 μL SYBR mix, 1 μL cDNA, 0.2 μL forward and reverse primers (10 μmol/L), and 3.6 μL RNase free water. Quantitative real-time PCR was performed on Bio-Rad CFX96 Real-Time PCR System (Bio-Rad, Hercules, CA, USA). The expression of the genes was calculated relative to the expression of the housekeeping gene β-actin with the formula 2−ΔΔCt. Amplification of specific transcripts was confirmed by melting-curve profiles at the end of each PCR. The primer sequences are listed in Appendix A.

### 2.6. Bioinformatics and Statistical Analysis

All data were presented as the mean ± SEM. Comparing the differences of non-normally various indicators among groups was conducted using by the Kruskal–Wallis one-way ANOVA with the Benjamini–Hochberg false discovery rate multiple-testing correction. The differences were considered significant when *p* < 0.05 and a trend when the *p* value was between 0.05 and 0.10. Correlations between intestinal microbiota and SCFAs were analyzed using Spearman’s correlation analysis with the “pheatmap” package in R (v4.1.0).

## 3. Results

### 3.1. Growth Performance Was Not Adversely Affected by Dietary Fiber Level

The ADG of pigs in group A, B and C was 0.36 ± 0.10 kg/d, 0.31 ± 0.08 kg/d and 0.32 ± 0.11 kg/d, respectively. The feed conversion rate in each group was 4.72 ± 0.61, 4.94 ± 0.65 and 5.08 ± 0.73, respectively. No significant differences were observed among groups in the growth performance of pigs (Table 2).

### 3.2. Fiber Apparent Digestibility Significantly Improved with Dietary Fiber Increasing

The ATTD of CP and CF were determined (Table 3). As dietary fiber increased, no significant differences in the ATTD of CP (one-way ANOVA, *p* = 0.72) was observed among group A (CP: 0.86 ± 0.01), group B (CP: 0.87 ± 0.03), and group C (CP: 0.87 ± 0.02). However, a high-fiber diet significantly increased the ATTD of CF, with values of 0.59 ± 0.07, 0.73 ± 0.03, and 0.76 ± 0.01 for groups A, B, and C, respectively (*p* < 0.05).

### 3.3. High Fiber Level Altered the Diversity of Cecal and Colonic Microbiota

The cecal and colonic microbial structure and composition were revealed by 16S rRNA gene sequencing. A total of 801,735 and 1,008,607 high-quality sequences were generated from cecal and colonic samples, respectively, with an average of 53,449 and 67,240 sequences per sample. The coverage index in this experiment consistently exceeded 0.99, indicating that the sequencing results adequately represent the breadth and depth of the samples. There were no significant differences in species richness (as indicated by the ACE and Chao1 indices) or diversity (as indicated by Shannon and Simpson indices) in the cecal and colonic microbiota at the taxonomic level (Figure 1a,b). However, ANOISM analysis based on Bray–Curtis distance showed that differences in the cecal and colonic microbiota among groups were significantly greater than differences within each group (*p* < 0.05) (Figure 1c,d). This suggested notable variations in the composition of cecal and colonic microbiota in Tunchang pigs under different dietary fiber levels (*p* < 0.05).

### 3.4. High Fiber Level Increased the Relative Abundance of Intestinal Microbiota Related to Fiber Degradation

In the cecum, Firmicutes, Bacteroidetes and Proteobacteria were the three dominant phyla (Figure 2a), which accounted for 51.30%, 36.61%, and 6.77% in group A, 42.74%, 46.62%, and 8.13% in group B, and 43.72%, 48.22% and 5.65% in group C, respectively. The relative abundance of Bacteroidetes significantly increased (*p* < 0.05). It is worth noting that there was a significant decreasing trend (*p* < 0.10) in the relative abundance of Firmicutes and Verrucomicrobia with the increase in dietary fiber levels. At the genus level, the 10 most predominant genera (those with a relative abundance ≥3% in at least one group) were Prevotella, unidentified_Bacteroidales, unidentified_Ruminococcaceae, unidentified_S24-7, unidentified_Lachnospiraceae, CF231, Phascolarcto-bacterium, Succinivibrio, and SMB53 (Figure 2b). The relative abundance of CF231 was significantly increased in group C compared with other groups (*p* < 0.05).

In the colon, similar to cecum, the three dominant phyla were Bacteroidetes, Firmicutes, and Proteobacteria (Figure 2a), contributing 46.54%, 44.38% and 4.47% to group A, 55.47%, 39.35% and 1.70% to group B, and 55.06%, 38.30% and 3.81% to group C, respectively. There was a significant increasing trend (*p* < 0.10) in the relative abundance of Bacteroidetes with the increase in dietary fiber levels. At the genus level, the relative abundance of Prevotella significantly increased in group B compared with group A (*p* < 0.05) (Figure 2b).

Some key microbial biomarkers were identified by linear discriminant analysis (LDA) coupled with LDA Effect Size. In the cecum, four microbial biomarkers (LDA scores > 2) were obtained from group B and C, including f_Clostridiaceae, g_RFN20, s_unidentified RFN20, and s_[Eubacterium] biforme (Figure 2c). In the colon, there were seven microbial biomarkers in group B and C, mainly from the Prevotella genus, such as *Prevotella stercorea*, as well as f_Clostridiaceae and g_Collinsella (Figure 2d). These microbial colonies were likely to be associated with a fiber degradation function.

### 3.5. Dietary Fiber Promotes the Increase in the Functionality Activity of the Microbiota and SCFAs

The alpha diversity analysis and LDA focused on the diversity and composition of intestinal microbiota. Microbial functions of different microbiota among groups were predicted with PICRUSt2 analysis, referring to the MetaCyc and KEGG database. The PICRUSt2 analysis showed that most microbial metabolic pathways were enriched in biosynthesis, degradation, generation of precursor metabolite and energy processes. In terms of biosynthesis, the main focus was on amino acids, cofactors, coenzymes, electron carriers, vitamins, nucleosides and nucleotides, as well as fatty acid and lipid biosynthesis. In terms of degradation, the main focus was on carbohydrate, carboxylate, nucleoside and nucleotide degradation. The most significant processes in the generation of precursor metabolite and energy was fermentation, followed by glycolysis and the tricarboxylic acid cycle (Figure 3a). Similar results were found in the colon. One-way ANOVA analysis found several microbial pathways of KEGG and MetaCyc had higher abundance in group B and C compared to those in group A (*p* < 0.05), such as fructose and mannose metabolism, lipopolysaccharide biosynthesis, and lipid IVA biosynthesis (Figure 3b,c). These enriched microbial pathways helped us understand the active metabolism in cecal and colonic microbiota, and also demonstrated that these microbiota ferment undigested carbohydrates that are not absorbed in the small intestine to produce SCFAs and provide energy.

The SCFA concentrations in the cecum and colon were next determined using GC-MS. One-way ANOVA revealed that the cecal concentrations of isobutyrate, valerate and isovalerate significantly increased (*p* < 0.05) in group C compared with other groups (Figure 3d). Dietary fiber level did not affect the acetate, propionate and butyrate concentrations in the cecum. The colonic SCFA concentrations were not affected by dietary fiber level.

### 3.6. Differentially Expressed Genes Related to Roughage Tolerance in the Cecum

Transcriptome sequencing was employed to examine the impact of dietary fiber levels on mRNA expression of genes in the cecum. Enrichment analysis of DEGs revealed that most of these genes are enriched in pathways related to cell signaling and cell adhesion (Figure 4a). Notably, the mRNA expressions of PYY and AQP8 were significantly increased in the cecum of pigs in group B compared with group A, whereas the mRNA expressions of CKM and CNN1 were significantly decreased (*p* < 0.05) (Figure 4b). Additionally, The mRNA expression of SLC5A8 was significantly elevated in the cecum of pigs fed high-fiber diets (*p* < 0.05). The results of quantitative real-time PCR corroborated the findings from the transcriptome analysis (Figure 4c).

## 4. Discussion

Dietary fiber is essential for the normal physiological metabolism of monogastric animals, as revealed by studies on its physicochemical properties and nutritional functions [21]. Our results showed no significant difference in the growth performance of pigs fed high-fiber diets, indicating that Tunchang fattening pigs could tolerate the diet with 5% CF. Similar results have been reported previously [9,22,23]. However, there were some contradictory results in the previous studies. For example, the inclusion rate of distiller dried grains with solubles was increased from 0 to 20% in nursery pig diets [24] and from 0 to 30% in growing-pig diets [25]. This resulted in a linear decrease in pig body weight, despite the diets being formulated to contain similar net energy. Various factors, such as genetic background and husbandry management can lead to different outcomes in similar studies, both of which affect growth performance. Tunchang pigs have gradually developed good performance on roughage tolerance through long-term breeding [12].

The roughage tolerance of Tunchang pigs was demonstrated by the significant improvement in the ATTD of CF with increased dietary fiber. The ATTD determined using AIA as an indigestible marker was highly correlated with that determined using total fecal collection [26], providing a more intuitive understanding of the digestion and absorption of nutrients in pigs. Compared with other animal trials [27,28], the ATTD of CP and CF determined in the present study were slightly higher. We speculated that pigs ingested a small amount of dust ash during feeding and drinking, leading to higher AIA content in feces. Given the small within-group error, the ATTD reflected the nutrient digestion and absorption in pigs. 

Due to pigs’ inability to secrete fiber-digesting enzymes, dietary fiber degradation primarily relies on fermentation by the cecal and colonic microbiota. Studies have shown that diet plays a crucial role in regulating the composition of intestinal microbiota [29], with dietary fiber being a key driver of these changes [30]. Changes in dietary fiber can alter the intestinal microbiota’s metabolic pathways and reorganize the microbial community in order to adapt to the new diet. In this study, high-fiber diets significantly increased the relative abundance of Bacteroidetes in the cecum, with a significant increasing trend in the colon. Bacteroidetes are dominant symbiotic bacteria in the intestines, with a sophisticated capacity to degrade polysaccharides and extract energy [31]. Additionally, Xie et al. [32] speculated that the higher abundance of Bacteroidetes in the intestines of Laiwu pigs may be related to the excellent roughage tolerance of Chinese indigenous pig breeds. A significant decreasing trend was observed in the relative abundance of Verrucomicrobia in cecal and colonic microbiota with increased dietary fiber. Similarly, a previous study reported a significant decrease in the relative abundance of Verrucomicrobia when Heigai pigs were fed a diet containing 30% alfalfa [33]. At the genus level, *Prevotella* was consistently one of the dominant genera, regardless of the dietary fiber level and intestinal segment. This finding is in line with previous studies [34,35,36], indicating that *Prevotella* may be a signature genus in intestinal microbiota of pigs. Meanwhile, the relative abundance of *Prevotella* in the colon significantly increased with the increase in dietary fiber. As a genus of Bacteroidetes, *Prevotella* exhibits diverse capabilities in utilizing complex carbohydrates [37]. In the cecum, *CF231* had a high relative abundance, and its relative abundance significantly increased with the increase in dietary fiber, especially in group C, accounting for 11.05%. The *CF231* is commonly present in the rumen of ruminants, and is one of the dominant genera [38,39]. Huang et al. [40] found a significant positive correlation between *CF231* and carbohydrate metabolism in the functional prediction of rumen microbiota in dairy cows. In a study on the differences of colonic microbiota and microbial metabolism between Landrace and Meishan pigs, correlation analysis showed a significant positive correlation between *CF231* and SCFAs [41]. 

To understand the differences in intestinal microbiota associated with high or low ATTD of CF, LDA was used to identify potential microbial biomarkers related to roughage tolerance, such as *Prevotella stercorea* and *Eubacterium biforme*. The PICRUSt2 accurately predicts the functional potential of microbial communities based on 16S rRNA sequencing data [42]. In this study, PICRUSt2 analysis revealed that the cecal and colonic microbiota primarily ferment CF to produce SCFAs for energy. This underscores the crucial role of intestinal microbiota in the host’s roughage tolerance. There were higher cecal concentrations of isobutyrate, valerate and isovalerate in group C compared with the other two groups, while there was no significant difference in the colonic SCFA concentrations among groups. Subsequent correlation analysis revealed the main relationships between intestinal microbiota and SCFAs, explaining the significant differences in the cecal SCFA concentrations.

SCFAs are an important source of energy for the host and also regulate host gene expression, thereby modulating physiological activities [43]. Transcriptome sequencing was used to identify DEGs in the cecum of Tunchang pigs among three groups with different dietary fiber level. It could help us further explore candidate genes and pathways that related to roughage tolerance of Tunchang pigs. In this study, enrichment analysis of DEGs revealed that most of these genes are involved in pathways related to cell signaling and cell adhesion. Notably, the mRNA expression of PYY was significantly increased in the cecum of pigs in group B. PYY encodes peptide YY, a short peptide released by intestinal cells that regulate satiety. By binding to neuropeptide Y receptors, peptide YY inhibits gastrointestinal motility and increases water absorption [44]. Correspondingly, there was significant downregulation of CKM and CNN1, which are involved in regulating smooth muscle contraction, and significant upregulation of AQP8, encoding aquaporin proteins. Furthermore, a recent study indicated that peptide YY also functions as an antimicrobial peptide, capable of inhibiting the transformation of virulent Candida albicans and selectively promoting the proliferation of non-pathogenic yeast forms. It plays a crucial role in maintaining fungal balance in the mammalian digestive system [45]. Additionally, the mRNA expression of SLC5A8 was significantly increased in the cecum and colon of pigs fed with high-fiber diets. SLC5A8, also known as SMCT1, encodes sodium-coupled monocarboxylate transporter 1, which mediates the active transport of SCFAs within the intestine, facilitating their absorption by intestinal epithelial cells [46]. Meanwhile, a study in mice revealed that SLC5A8 inhibits the differentiation of naive T cells into pro-inflammatory IFN γ-secreting cells by transporting butyrate. It has also been observed that SLC5A8 can prevent colitis and colon cancer under conditions of low fiber intake. It is concluded that SLC5A8 serves as a crucial link between dietary fiber and the mucosal immune system through the bacterial metabolite butyrate [47]. Similarly, the upregulation of PYY and SLC5A8 was also shown in the cecum of pigs fed 15% dietary alfalfa meal [48]. We speculated that appropriate high-fiber diets could inhibit intestine motility and increase the absorption of water and SCFAs.

## 5. Conclusions

In conclusion, this study observed that Tunchang pigs can adapt to a high-fiber diet with 5% crude fiber. The adaptative alternation of intestinal microbiota and regulation of genes jointly promoted the host to digest appropriate high-fiber diets. These findings draw insight into the roughage tolerance of Tunchang pigs, and help in further exploring host–microbiota interactions.

## Figures and Tables

**Figure 1 animals-14-02076-f001:**
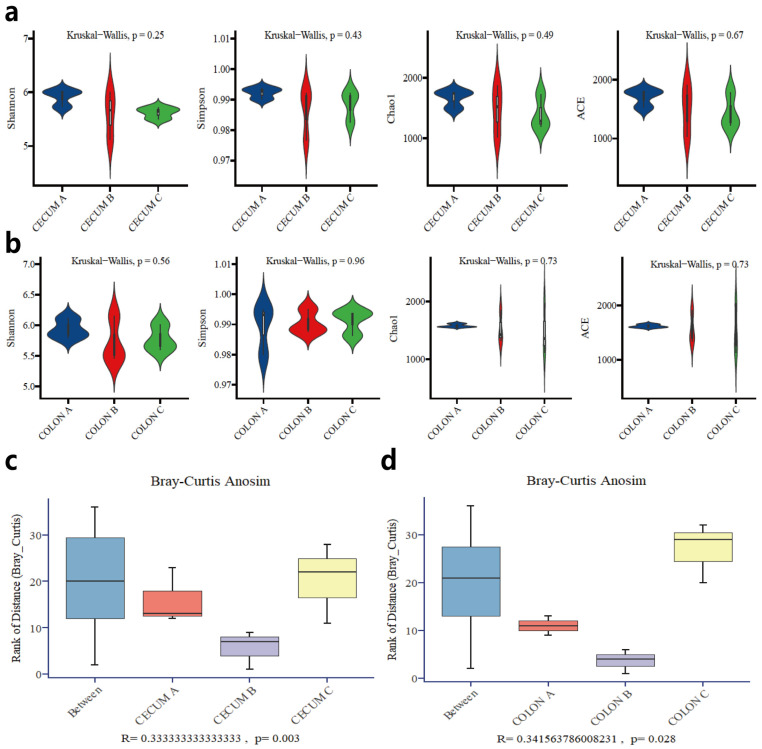
The comparison of microbial diversity of Tunchang pigs among groups. Alpha diversity of cecal microbiota (**a**) and colonic microbiota (**b**). Anoism test of cecal microbiota (**c**) and colonic microbiota (**d**) based on Bray–Curtis distance.

**Figure 2 animals-14-02076-f002:**
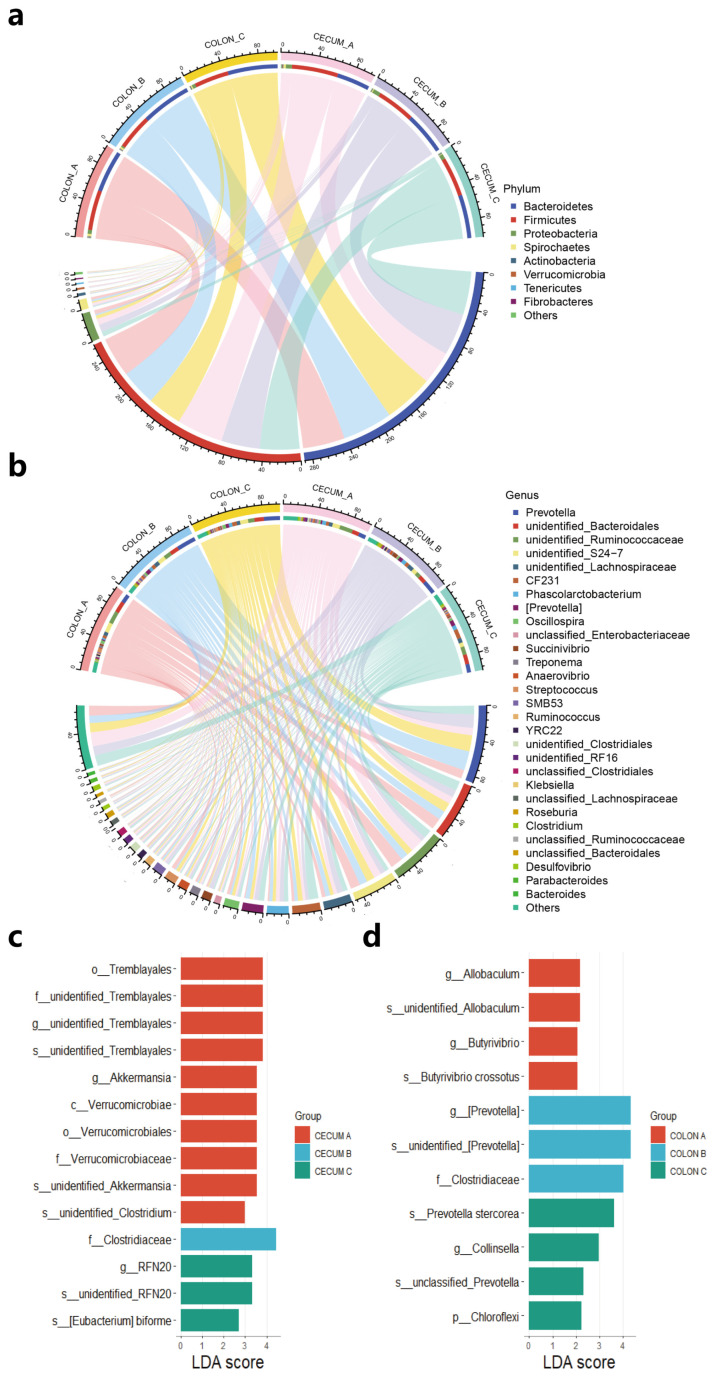
The intestinal microbial composition in the cecum and colon of Tunchang pigs evaluated at the phylum (**a**) and genus (**b**) level. The LDA analysis in the cecum (**c**) and colon (**d**).

**Figure 3 animals-14-02076-f003:**
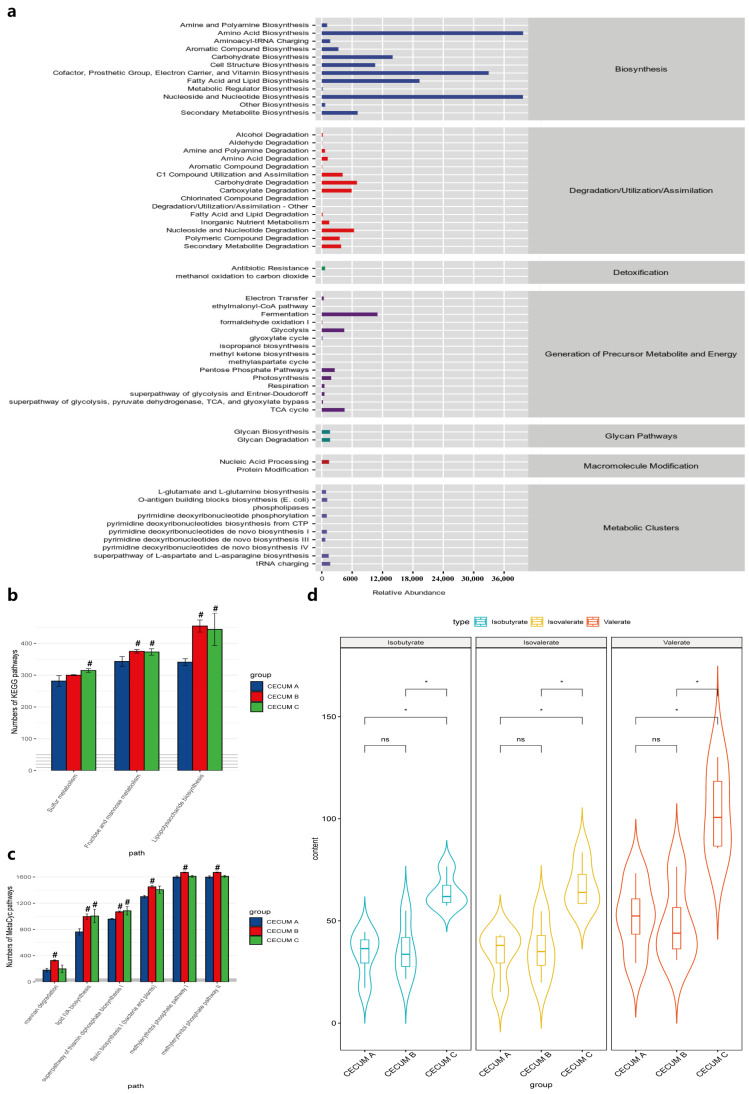
The predicted functions of cecal microbiota of Tunchang pigs based on PICRUSt analysis (**a**). Significantly changed microbial functions of KEGG (**b**) and MetaCyc (**c**). ^#^
*p* < 0.05 compared with group A. Significantly changed SCFAs (**d**). * *p* < 0.05; ns, not significant.

**Figure 4 animals-14-02076-f004:**
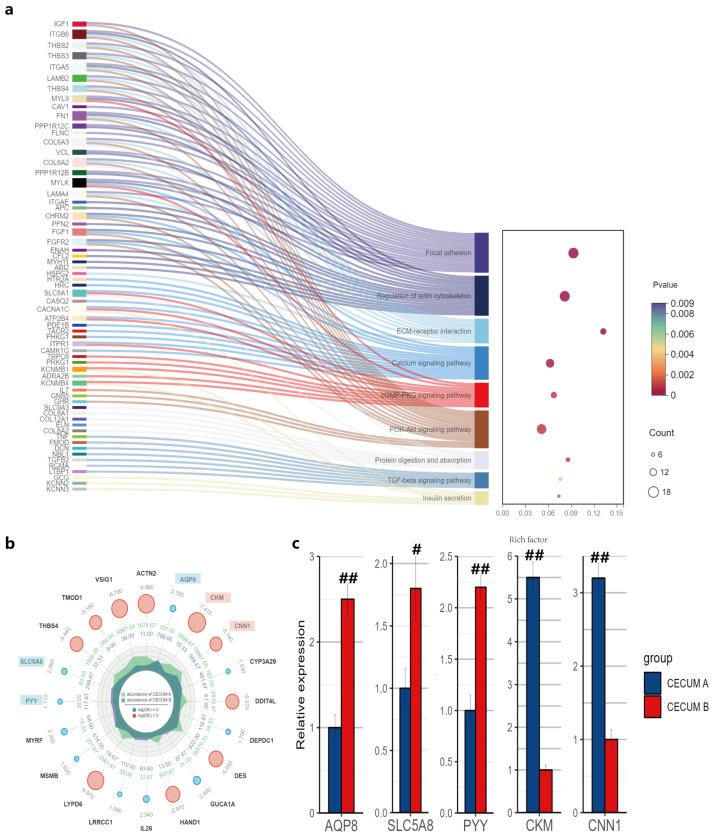
The top 20 differentially expressed genes between group A and B (**a**) in the cecum of Tunchang pigs. KEGG enrichment analysis of differentially expressed genes (**b**). Relative expression of several genes as determined by quantitative real-time PCR (**c**). AQP8, aquaporin 8; SLC5A8, solute carrier family 5 member 8; PYY, peptide YY; CKM, creatine kinase, M-type; CNN1, calponin 1. ^#^
*p* < 0.05, ^##^
*p* < 0.01.

**Table 1 animals-14-02076-t001:** Ingredients and nutrient level of the experimental diets (as-fed basis).

Items	Group A	Group B	Group C
Ingredient (%)			
Corn	60.95	56.50	52.00
Soybean meal	28.00	25.30	24.10
Wheat bran ^1^	2.00	9.50	15.50
Grease meal	3.00	2.80	2.60
Fish meal	2.00	1.85	1.75
Antifungal agent ^2^	0.05	0.05	0.05
Premix ^3^	4.00	4.00	4.00
Nutrient level ^4^			
CP (%)	16.54	16.36	16.20
CF (%)	3.57	4.34	5.00
DE (MJ/KG)	15.10	15.09	15.07

^1^ Wheat bran: 14.11% CP, 13.99% CF, 0.15% AIA. ^2^ Antifungal agent is a feed mold inhibitor that effectively prevents the harm of various mycotoxins to pigs. ^3^ Premix supplied per kg diet as follows: vitamin A 8000 IU, vitamin D_3_ 1500 IU, vitamin E 100 mg, vitamin K_3_ 4 mg, vitamin B_l_ 2 mg, vitamin B_2_ 8 mg, vitamin B_6_ 3 mg, vitamin B_12_ 0.04 mg, niacin 30 mg, pantothenie acid 35 mg, folic acid 0.6 mg, biotin 0.13 mg, Choline 150 mg, Fe 60 mg, Cu 5 mg, Zn 60 mg, Mn 10 mg, Se 0.15 mg and I 0.1 mg. ^4^ The DE is calculated, whereas all other values are analyzed.

**Table 2 animals-14-02076-t002:** Responses of growth performance of Tunchang pigs to dietary fiber levels.

Items	Group A	Group B	Group C	*p*-Value
Initial weight (kg)	41.63 ± 2.74	45.75 ± 1.10	43.75 ± 3.14	0.38
Final weight (kg)	82.81 ± 13.17	80.86 ± 9.56	80.44 ± 14.43	0.93
ADG (kg/d)	0.36 ± 0.10	0.31 ± 0.08	0.32 ± 0.11	0.60
ADFI (kg/d)	1.68 ± 0.34	1.51 ± 0.22	1.59 ± 0.35	0.60
F/G	4.72 ± 0.61	4.94 ± 0.65	5.08 ± 0.73	0.62

**Table 3 animals-14-02076-t003:** Responses of the ATTD of CP and CF to dietary fiber levels.

Items	Group A	Group B	Group C	*p*-Value
CP	0.86 ± 0.01	0.87 ± 0.03	0.87 ± 0.02	0.98
CF	0.59 ± 0.07 ^a^	0.73 ± 0.03 ^b^	0.76 ± 0.01 ^b^	0.00

Kruskal–Wallis one-way ANOVA was conducted; different letters in the same row indicate a signifi- cant difference (*p* < 0.05), while the lack of variation in letters indicates no significant difference (*p* > 0.05).

## Data Availability

The datasets used or analyzed during the current study are available from the corresponding author on reasonable request.

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
