# Peer review of "The Adaptive Alternation of Intestinal Microbiota and Regulation of Host Genes Jointly Promote Pigs to Digest Appropriate High-Fiber Diets"

_animals, 2024, doi:10.3390/ani14142076_

Round 1

Reviewer 1 Report

Comments and Suggestions for Authors

The study is interesting as it seeks to delve into the world of fiber, a group of dietary components that is still little explored in the nutrition of monogastric animals. This is so true that the term crude fiber is still used, which does not differentiate the diversity of fiber components, which can have different impacts on the gastrointestinal tract.

It is well known that dietary fiber does affect the microbiome composition in the intestinal tract of animals, since fiber is a substrate for bacterial growth in the hindgut and fermentation increasing volatile fatty acid production. Still, the study makes an interesting contribution as it deepens the detailed assessment of the microbiota as well as the expression of some genes.

Questions:

- Table 1: How is it possible a wheat bran contain 14% CF and at the same time 9% EE? Was there added an external fat to this feed ingredient?

- How can we explain that animals in group C, compared to those in group A, showed better CF digestion and (numerically) better EE digestion (Table 3), but feed conversion tended to be worse for group C (Table 2)?

The average age of the animals at the beginning of the experiment is not stated, only the duration of the experiment (111 days) is stated. It is important to add this information. As shown in Table 2, the performance of the animals was not good, which may partially be due to the genetic material. Higher performance animals would possibly be more sensitive to different diets. This is one of the reasons why the performance result should be taken with caution. An additional problem is the difference of almost 10% in the average body weight of the animals between groups A and B at the beginning of the experiment (Table 2), which indicates high variation between animals and, consequently, low sensitivity of the experiment. So much so that it was not possible to detect a significant difference of 2 kg in final body weight between treatments, nor the considerable difference between the conversions of 4.72 vs 5.08. The animal performance part is definitely a weak point of the study. The strong point lies in the evaluation and discussion of the composition of the microbiota as well as gene expression.

Author Response

Comments 1: How is it possible a wheat bran contain 14% CF and at the same time 9% EE? Was there added an external fat to this feed ingredient?

Response 1:  Thanks for the kind and insightful comments. We carefully checked the data and referred to the literatures, finding that wheat bran contain approximately 3.5% ether etract.  Due to the lack of some key instruments in our laboratory, the EE content was measured by Personal Biotechnology Co., Ltd. Given the time and materials constraints, we decided to remove all EE data from the manuscript, which would not affect the study conclusion. We apologize for our oversight and once again thank you for your useful and valuable comments.

Comments 2: How can we explain that animals in group C, compared to those in group A, showed better CF digestion and (numerically) better EE digestion (Table 3), but feed conversion tended to be worse for group C (Table 2)?

Response 2: Thanks for the kind and insightful comments. We think this result can be attributed to two reasons: first, the addition of wheat bran slightly reduced the CP and DE in the feed, necessitating an increase in digestibility and intake by the pigs to obtain sufficient nutrients; second, one-way ANOVA indicated there was no significant difference in these data.

Comments 3: The average age of the animals at the beginning of the experiment is not stated, only the duration of the experiment (111 days) is stated. It is important to add this information. As shown in Table 2, the performance of the animals was not good, which may partially be due to the genetic material. Higher performance animals would possibly be more sensitive to different diets. This is one of the reasons why the performance result should be taken with caution. An additional problem is the difference of almost 10% in the average body weight of the animals between groups A and B at the beginning of the experiment (Table 2), which indicates high variation between animals and, consequently, low sensitivity of the experiment. So much so that it was not possible to detect a significant difference of 2 kg in final body weight between treatments, nor the considerable difference between the conversions of 4.72 vs 5.08. The animal performance part is definitely a weak point of the study. The strong point lies in the evaluation and discussion of the composition of the microbiota as well as gene expression.

Response 3: Thanks for the kind and insightful comments and suggestion. The average age of pigs is 180 days, which is added in L70. In this study, We selected Tunchang pigs as the experimental animals to verify their roughage tolerance and investigate the underlying mechanism. However, due to their limited population, we could not find 24 Tunchang pigs with nearly identical body weights, which introduced some errors into the performance result. This is a weak point of our study.

These comments are all valuable and very helpful for revising and improving our paper, as well as the important guiding significance to our researches. Thank you very much for your attention.

Reviewer 2 Report

Comments and Suggestions for Authors

This paper is interesting in describing the alterations which occur if a high fiber diet is fed.

The title of the paper suggests that the regulation of host genes promote the digestion of high fiber diets. In my opinion the intestinal microbiota alteration is caused by the kind of substrate that is offered to the microflora and the products of that fermentation affects the host genes.

Regarding the experimental diets: 

The wheat composition mentions an EE% of 9.22%. either this number is not correct of the supplement was not wheat bran. normally wheat brancontains around 3,5 % EE. Can the authors explain what happened. 

Author Response

Comments 1: The title of the paper suggests that the regulation of host genes promote the digestion of high fiber diets. In my opinion the intestinal microbiota alteration is caused by the kind of substrate that is offered to the microflora and the products of that fermentation affects the host genes.

Response 1: Thank you for your insightful comments on our paper. Your observations highlight a crucial aspect of the complex interplay between diet, gut microbiota, and host gene expression. We agree that the type of substrate provided to the intestinal microbiota significantly influences its composition and activity. The metabolites produced by microbial fermentation indeed interact with host cells, leading to changes in gene expression that promote the digestion of high-fiber diets. Studies indicated that the weight and volume of the colon change with dietary fiber. It is still hard to say whether the host will self-regulate to adapt to a high-fiber diet. We appreciate your comments and have incorporated this perspective to further clarify our discussion on the regulation of host genes.

Comments 2: The wheat composition mentions an EE% of 9.22%. either this number is not correct of the supplement was not wheat bran. normally wheat brancontains around 3,5 % EE. Can the authors explain what happened.

Response 2: Thanks for the kind and insightful comments. We carefully checked the data and referred to the literatures, finding that wheat bran contain approximately 3.5% ether extract.  Due to the lack of some key instruments in our laboratory, the EE content was measured by Personal Biotechnology Co., Ltd. Given the time and materials constraints, we decided to remove all EE data from the manuscript, which would not affect the experimental conclusion. We apologize for our oversight and once again thank you for your useful and valuable comments.

Reviewer 3 Report

Comments and Suggestions for Authors

This is an interesting study, in which authors investigated the effects of high-fiber diets on Tunchang pigs, focusing on growth performance, fiber digestibility, intestinal microbiota, short-chain fatty acids (SCFAs) concentrations, and intestinal gene regulation. Take-home message was that high fiber diets altered intestinal microbiota diversity, increasing the abundance of certain bacteria like Prevotella, and SCFAs such as isobutyrate, valerate, and isovalerate. Potential microbial biomarkers for roughage tolerance were identified, including Prevotella stercorea and Eubacterium biforme. Also, high-fiber diets influenced gene expression by upregulating PYY, AQP8, and SLC5A8, which may enhance water and SCFA absorption, and downregulating CKM and CNN1, possibly inhibiting intestinal motility.

The work is commendable; however, it is extremely difficult to provide an accurate revision at its present form. I would urge the authors to re-check the entire manuscript for English accuracy, spelling errors and writing. Many sentences are completely disconnected or have serious errors, some examples provided below:

L39: Despite these studies have 39 revealed the significant impact of dietary fiber on growth performance and nutrient di- 40 gestibility, there are some limitations without considering different pig breeds and appro- 41 priate dietary fiber level.

L45: Pigs are lack of digestive enzymes to degrade dietary fiber.

L272: Transcriptome sequencing was used to analysis and screen the DEGs in cecum.

L341: The SCFAs are an important source of energy for the host, and these metabolites also 341 participate in regulating host gene expression, thereby modulating physiological activities

L353: Further- 353 more, a recent study has indicated that peptide YY also functions as an antimicrobial pep- 354 tide, capable of inhibiting the transformation of more virulent Candida albicans and se- 355 lectively promoting the proliferation of non-pathogenic yeast forms, which plays an im- 356 portant role in maintaining fungal balance in the mammalian digestive system [45].

I would be happy to reassess the paper after English revision.

Comments on the Quality of English Language

it is extremely difficult to provide an accurate revision at its present form. I would urge the authors to re-check the entire manuscript for English accuracy, spelling errors and writing. Many sentences are completely disconnected or have serious errors, some examples provided below:

L39: Despite these studies have 39 revealed the significant impact of dietary fiber on growth performance and nutrient di- 40 gestibility, there are some limitations without considering different pig breeds and appro- 41 priate dietary fiber level.

L45: Pigs are lack of digestive enzymes to degrade dietary fiber.

L272: Transcriptome sequencing was used to analysis and screen the DEGs in cecum.

L341: The SCFAs are an important source of energy for the host, and these metabolites also 341 participate in regulating host gene expression, thereby modulating physiological activities

L353: Further- 353 more, a recent study has indicated that peptide YY also functions as an antimicrobial pep- 354 tide, capable of inhibiting the transformation of more virulent Candida albicans and se- 355 lectively promoting the proliferation of non-pathogenic yeast forms, which plays an im- 356 portant role in maintaining fungal balance in the mammalian digestive system [45].

Author Response

Comment 1: The work is commendable; however, it is extremely difficult to provide an accurate revision at its present form. I would urge the authors to re-check the entire manuscript for English accuracy, spelling errors and writing. Many sentences are completely disconnected or have serious errors, some examples provided below

Response 1: Thank you for your valuable and thoughtful comments. We have carefully checked and improved the English writing in the revised manuscript. Now the contexts have been presented in a better manner.  We hope it will meet  the English presentation standard. some examples provided below:

L40: These studies have revealed the significant impact of dietary fiber on growth performance and nutrient digestibility. However, there are certain limitations as they do not investigate the underlying interactions.

L45: The digestive enzymes secreted by the gastrointestinal tract of pig cannot degrade dietary fiber

L269: Transcriptome sequencing was utilized to analyze and identify the DEGs in the cecum

L337: SCFAs are an important source of energy for the host and also regulate host gene expression, thereby modulating physiological activities.

L349: Furthermore, a recent study indicated that peptide YY also functions as an antimicrobial peptide, capable of inhibiting the transformation of virulent Candida albicans and selectively promoting the proliferation of non-pathogenic yeast forms. It plays a crucial role in maintaining fungal balance in the mammalian digestive system.

Round 2

Reviewer 3 Report

Comments and Suggestions for Authors

In the present study, Tunchang pigs were fed diets with varying levels of wheat bran (high fiber), which did not significantly impact their growth performance but significantly improved the apparent total tract digestibility of crude fiber. The diets also altered the diversity of intestinal microbiota AND modulated the production of BCVFA. Potential microbial biomarkers associated with roughage tolerance were also identified. The work has merit, but some adjustments are still required.

L12-15: please rephrase this section.

L22: fed diets with different wheat..

L25: with increasing dietary fiber content.

L28: appropriate

L30: Tt?

L37: to the utilization of agricultural

L39-40: this section still does not make sense.

L60: green feed? Is this correct?

L62-66: Please state your objectives first, and then your hypothesis. I would suggest:

“The present study aimed to investigate the interaction between intestinal microbiota and gene expression in response to feeding high-fiber diets to Tunchang pigs through 16S rRNA gene sequencing and transcriptome sequencing. It was hypothesized that Tunchang pigs would perform well in roughage tolerance and that dietary fiber would affect the composition and function of cecal and colonic microbiota, involving the regulation of intestinal gene expression.”

L76: adjusted by manipulating the wheat bran content.

L78: Could the authors explain why diets were formulated on a DE and not ME basis?

L91: Nutrient and fiber digestibility

L91: Please indicate the selection criteria for the 5 pigs/treatment.

L147: No significant differences were observed..

L152: Fiber is not a nutrient – please adjust here and elsewhere.

L162: Please be consistent with the use of level vs. concentration vs. content throughout the manuscript.

L180: It is worth noting..

L202: “The above analysis..” Which analysis?

L212: “several microbial pathways..” Could you please exemplify?

L260: Fiber is not a nutrient. Please adjust here and elsewhere.

L265-276: Please provide some brief sentences contrasting your findings with the literature.

L288: Reference?

L314: This finding is in line with previous studies [34-36]..

L347: What is the practical relevance of PYY modulation?

Comments on the Quality of English Language

NA

Author Response

Comments 1: L60: green feed? Is this correct?

Response 1:  Thanks for the kind and insightful comments and suggestion. Green feed is a correct term referring to fresh, green plant material used as feed for livestock. This can include a variety of plants such as grasses, legumes, and other forages that are harvested while still green and rich in nutrients.

Comments 2: L78: Could the authors explain why diets were formulated on a DE and not ME basis?

Response 2: Thanks for the kind and insightful comments and suggestion. DE values are often easier to determine experimentally compared to ME values, which require more complex measurements of energy losses such as through urine and gases. Therefore, DE-based formulations can be more practical for day-to-day feeding management.

Comments 3: Please indicate the selection criteria for the 5 pigs/treatment.

Response 3: Thanks for the kind and insightful comments and suggestion. The population of Tunchang pigs is small. To fulfill the national preservation task, five pigs were randomly selected from each of the three groups for slaughter. Randomization ensures that each treatment group is as similar as possible at the start of the experiment.

Comments 4: Please be consistent with the use of level vs. concentration vs. content throughout the manuscript.

Response 4: Thanks for the kind and insightful comments and suggestion. "Level" is referring to the amount or degree of a variable within a broader context. "Concentration" is referring to the amount of a substance within a specific volume or mass of another substance. "Content" is referring to the absolute amount of a substance in a given quantity of material, typically expressed as a percentage or in specific units. We think that these three words should be used distinctively.

Comments 4: L202: “The above analysis..” Which analysis?

Response 4: We are sorry for our overlook. The above analysis include alpha diversity analysis and LDA, which are added in L202.

Comments 5: L212: “several microbial pathways..” Could you please exemplify?

Response 5: We are sorry for our overlook. Several microbial pathways include fructose and mannose metabolism, lipopolysaccharide biosynthesis, lipid IVA biosynthesis, which are added in L212.

Comments 6: L265-276: Please provide some brief sentences contrasting your findings with the literature.

Response 6: Thanks for the kind and insightful comments and suggestion. We think this paragraph is repetitive with the preceding and following text, making it excessively long, so we deleted it.

Comments 7: Reference?

Response 7: We are sorry for our overlook. The reference has been supplemented.

Comments 8: What is the practical relevance of PYY modulation?

Response 8: Thanks for the kind and insightful comments and suggestion. PYY is a hormone that plays a significant role in regulating appetite and food intake. the practical relevance of PYY modulation lies in its potential to influence eating behaviors, manage metabolic diseases, and improve gastrointestinal health.

These comments are all valuable and very helpful for revising and improving our paper, as well as the important guiding significance to our researches. Thank you very much for your attention.